# Ant parasitoidism in checkered beetles: *Phyllobaenus obscurus* developing inside intact cocoons of two species of the *Ectatomma ruidum* species complex

Gabriela Pérez-Lachaud[1]*, Chantal Poteaux[2], John M. Leavengood Jr[3], Jean-Paul Lachaud[1]*

1 Dpto. Conservación de la Biodiversidad, El Colegio de la Frontera Sur, Chetumal, Quintana Roo, Mexico, 2 Laboratoire d'Éthologie Expérimentale et Comparée UR 4443 (LEEC), Université Sorbonne Paris Nord, Villetaneuse, France, 3 United States Department of Agriculture, APHIS, PPQ, Tampa, Florida, United States of America

* igperez@ecosur.mx (GP-L); jlachaud@ecosur.mx (J-PL)

## Abstract

Known parasitoids of ants include species from several families of flies, wasps, strepsipterans, nematodes, and mites. Curiously, while myrmecophily is heavily biased towards Coleoptera, one of the most diverse and speciose insect orders, no beetles specialized as parasitoids of ants have been recorded, although the parasitoid habit has evolved at least 13 times within this order. Here we report on observations that strongly suggest that a checkered beetle species behaves as a parasitoid of ant brood. A total of 146 colonies or part of colonies of three species of the *Ectatomma ruidum* species complex (*E. ruidum* sp. 2, 3, 4) were excavated in several sites along the Pacific coastal plains of Oaxaca, Mexico, during three collecting campaigns (2015–2017). Overall, 11060 adults, 5795 cocoons and 2185 larvae were examined. Upon dissection, four intact, ethanol-preserved cocoons contained ant prepupae/ pupae parasitized by characteristic campodeiform beetle larvae (prognathous head, three pairs of segmented legs, no prolegs, body with sparse but long pubescence), and a fifth cocoon presented a round exit hole. Sequencing of the COI gene fragment support assignment of the larva to Cleridae, consistent with the genus *Phyllobaenus* (Cleridae). An active, pink-colored larva that emerged from a cocoon containing the remains of the ant host pupa, was reared to adulthood and could be identified as *Phyllobaenus obscurus* (Gorham). Beetle larvae were found inside intact cocoons of two species: *E. ruidum* sp. 3 and sp. 4. The prevalence of parasitism is extremely low, representing less than 0.6% of available cocoons. Predatory during both adult and larval stages, checkered beetles are broadly known as predators of wood-boring and cone-boring beetles, and some species are facultative parasitoids of solitary bees or wasps or, very rarely, specialized in predating social insects. We assert that

**Data availability statement:** All relevant data are within the paper and its Supporting information files.

**Funding:** This study was partially funded by the project number M12A01, Ecos-Nord-CONACYT Program granted to CP and J-PL. The funders had no role in study design, data collection and analysis, decision to publish, or preparation of the manuscript.

**Competing interests:** The authors have declared that no competing interests exist.

the novel discovery of clerid-ant brood parasitoidism within the subterranean host colony deviates yet further from any adaptation known to date among clerids.

## Introduction

Many invertebrates have adopted a predatory or parasitoid life strategy and play an important role in biological control [1–3]. While predators capture and consume multiple prey during their developmental and/or adult life stage, a parasitoid is an organism in which the immature stages develop on or within a single organism (the host), with this feeding activity eventually resulting in the death or sterilization of the host [4,5], whereas adult parasitoid organisms are free-living. A diverse array of parasitoids specializes in attacking adult ants or their brood. Known ant parasitoids include species of several insect orders and families: flies, twisted-wing parasites, and parasitic wasps [6–11], two families of Nematoda [12], and one mite family [13–15], but no coleopteran species has been recorded as parasitoid of ants yet. Ant parasitoids are highly specialized symbionts that display a variety of morphological, chemical, and behavioral strategies for accessing and successfully parasitizing their social hosts [13,16,17]. For example, species in the genera *Elasmosoma* Ruthe and *Neoneurus* Haliday (Hymenoptera, Braconidae) that parasitize adult ants outside the nest have morphological adaptations for grasping workers and securing oviposition (e.g., tarsal modifications, raptorial legs) and rely on rapidity to oviposit directly into the ant body [18]. Wasps in the Eucharitidae family lay eggs away from their hosts, in or on plant substrate; they possess a very mobile first-instar larva, termed planidium, which is responsible for accessing the hosts through phoresis on foraging ants or attached to prey [19–23]. Chemical volatiles of glandular origin used by ants as pheromones are co-opted by ant-decapitating flies (Phoridae) to locate potential hosts [24]. Once in the vicinity, phorid females rely on short-range chemical signals, such as cuticular hydrocarbons, to locate and select potential hosts [25].

According to Eggleton and Belshaw [4,26], the parasitoid habit has evolved independently at least 13 times in Coleoptera. Beetles do not have piercing ovipositors and the poor flying ability of the adult contrasts with the mobility of the larval stages [4,26]. As in the case of eucharitid wasps and Strepsiptera, known species of Coleoptera with a parasitoid lifestyle lay eggs away from the hosts and possess phoretic first-instar larvae, termed triungulin, that are responsible for locating and selecting suitable hosts (Meloidae, Ripiphoridae, some Cleridae [4,19]). The phoretic larvae attach to the adults of their hosts either on living plants, dead wood or soil. Most parasitoid coleopteran families appear to have evolved from ancestors associated with dead wood, with some shifting to attacking insects in the soil. Hosts of coleopteran parasitoids include Diplopoda, Thysanura, Blattaria, Thysanoptera, other Coleoptera, Hymenoptera (other than Formicidae), and Lepidoptera [4]. Compared with parasitoid wasps and flies, their host taxonomic diversity is rather low, suggesting that other factors may constrain the host range of coleopteran parasitoids [4]. Contradictorily, evolution of myrmecophily in insects is heavily biased towards the Coleoptera. The nature of colony exploitation by myrmecophilous beetles varies significantly, with

species ranging from scavengers and refuse dwellers to tolerated or highly integrated guests [27]. Some adult myrme-cophilous beetles, as in the genus *Cremastocheilus* Knoch [28] (Scarabaeidae) or in various Paussinae species [29] (Carabidae), are predatory on ant brood and exhibit various strategies to locate and exploit their prey within the ant host nest. Some species, such as *Myrmedonota xipe* Mathis & Eldredge (Staphylinidae), can even use specialized parasitoid host location cues to selectively prey on *Azteca sericeasur* Longino workers that have been previously parasitized by phorid flies [30]. However, to the best of our knowledge, no species of beetle has ever been recorded as developing as a parasitoid of adult ants or their brood. Here we report on observations that strongly suggest that a species of Cleridae develops as a parasitoid of ant brood.

Cleridae is a family of about 4710 described species (R. Gerstmeier, personal communication). Predatory during both adult and larval stages, these beetles are broadly known as predators of wood-boring and cone-boring beetles (e.g., bark beetles, longhorn beetles, powderpost beetles) and are often observed in the galleries of their prey [19,31]. However, several clerid genera are known to feed on gall-makers and their parasitoids (e.g., *Cymatodera* Gray, *Neohydnocera* Leavengood, *Phyllobaenus* Dejean, *Placopterus* Wolcott), on pollen as a primary or secondary protein source (some Clerinae, Korynetinae, Epiclininae), on grasshopper egg pods (*Trichodes* Herbst, *Aulicus* Spinola), on non-gallmaking cat-erpillars (e.g., *Aulicus, Phyllobaenus*), on insects associated with carrion and stored food products and their pests (*Necro-bia* Latreille), or have been reared from cells of bee and wasp larvae (e.g., *Cymatodera, Lecontella* Wolcott & Chapin, *Phyllobaenus, Placopterus, Trichodes*) [31,32]. Many species exhibit several different larval feeding strategies, which all may appear to be specialized behaviors (see Table 1). For example, the larvae of *Placopterus thoracicus* (Olivier), which is strongly associated with wood-boring beetles, have been found inside insect galls, and the adults have been reared from cells of crabronid and pemphredonid wasps [48]. Similarly, in *Neohydnocera longicollis* (Ziegler), formerly referred to as *Hydnocera longicollis* Ziegler and *Isohydnocera curtipennis* (Newman), adults have been reported emerging from the galls of three different hosts in three different insect orders (Diptera, Hymenoptera and Lepidoptera) [38,45,56,57]. On the other hand, various species, mainly from the genus *Trichodes*, develop as cleptoparasitoids (development at the expense of a single host organism by at least partial usurpation of its food supply, resulting in the death of the host [4]), and many other species have been reared from galls or nest cells of Hymenoptera while also exhibiting other predatory behaviors [31,32,68] (see Table 1). In general, however, the natural histories of most species in these genera are poorly documented.

In Europe, the association between *Trichodes umbellatarum* Olivier and the larvae of various bee species borders par-asitoidism and predation [70,71], some larvae developing upon a single host as facultative parasitoids while others need several hosts to complete development, acting as predators. Similar shifts between predation and facultative parasitoid-ism have also been reported in the New World for some species such as *Lecontella brunnea* (Spinola), *Trichodes ornatus douglasianus* White, *T. simulator* Horn, and *N. longicollis* [42,51,52,57] (Table 1). However, no clerid species associated with hymenopterans can compare to the unique case of *Tarsobaenus letourneauae* Leavengood, Pinkerton & Rifkind and *T. piper* Leavengood, Pinkerton & Rifkind which exhibit a remarkable symbiotic relationship with ants and their host plants [72]. Larvae of both *Tarsobaenus* species have been reported inhabiting the petiole chambers of the Piperaceae *Piper cenocladum* Casimir de Candolle, and *P. obliquum* Ruiz & Pavón, respectively, which host the myrmicine ant *Pheidole bicornis* Forel, and feeding either on the ant brood when ants are present or on the plant-produced food bodies when ants are absent; remarkably, translocation experiments showed that, in the absence of ants, the clerid larva induced food-body production too [73]. Although the evolution of parasitism of this ant-plant mutualism is intriguing, we assert that the novel discovery of clerid-ant brood parasitism within *Ectatomma* colonies is without precedent among all previously known adaptations in clerids. Here we report on the first recorded instance of a checkered beetle developing as a parasitoid of the ant brood which also represents the first case of a primary parasitoid of ants from the order Coleoptera. We also sum-marize all known rearing records among North American Cleridae and discuss the evolutionary transitions and diversifica-tion of larval feeding habits within Cleridae.

**Table 1. Summary of known rearing records among North American Cleridae and larval feeding mode on insects other than wood, stem and twig borers.**

| Clerid subfamily and tribe | Species | From galls induced by insects | From nests/cells of solitary bees or wasps | From nests/cells of social Hymenoptera | Other insect prey | Larval feeding mode | References |
|---|---|---|---|---|---|---|---|
| Tillinae: no tribal assignment | *Cymatodera* sp. | | Megachilidae Vespidae | | | Pr | [33,34] |
| | *Cymatodera ovipennis* LeConte | | Apidae Megachilidae | | | Pr | [35] |
| | *Cymatodera undulata* (Say) | Cynipidae[Hy] | Sphecidae | | Gall inquilines Chalcididae[Hy] Cynipidae[H] Eurytomidae[Hy] Pteromalidae[Hy] | Pr | [36–39] |
| | *Lecontella brunnea* (Spinola) | | Unident. bees Megachilidae Sphecidae Vespidae | | | Pr, Pa? | [40–42] |
| Clerinae: Clerini | *Aulicus terrestris* Linsley | | | | Acrididae[Or] | Pr | [43] |
| | *Enoclerus rosmarus* (Say) | Unident. insect Tephritidae[Di] | | | Gall inquilines | Pr | [44,45] |
| | *Enoclerus* sp. | | | | Tortricidae[Le] | Pr | [46] |
| | *Enoclerus zonatus* (Klug) | | | | *Blastobasidae[Le] *Nitidulidae[Le] *Prodoxidae[Le] | Pr | [47] |
| | *Placopterus thoracicus* (Olivier) | Cynipidae[Hy] | Crabronidae Pemphredonidae | | Gall inquilines | Pr | [37,48] |
| Clerinae: Dieropsini | *Trichodes apivorus* Germar** | | Unident. bees Unident. wasps | | | Pr | [49,50] |
| | *Trichodes bibalteatus* LeConte | | Megachilidae | | | Pr | [51] |
| | *Trichodes bicinctus* Green | | Apidae | | | Pr | [51] |
| | *Trichodes nuttalli* (Kirby)** | | Unident. bees  Unident. wasps | | Acrididae[Or] | Pr | [50,51] |
| | *Trichodes oregonensis* Barr | | | | Acrididae[Or] | | [51] |
| | *Trichodes oresterus* Wolcott | | Megachilidae | | | | [51] |
| | *Trichodes ornatus bonnevillensis* Foster | | Megachilidae Vespidae | | | Pr | [33,34,51] |
| | *Trichodes ornatus douglasianus* White | | Apidae Megachilidae Vespidae | | | Pr, Pa? | [51–54] |
| | *Trichodes ornatus hartwegianus* White | | Megachilidae | | | Pr | [51] |
| | *Trichodes ornatus ornatus* Say | | Unident. bees Unident. wasps Megachilidae | | Post-copulatory cannibalism | Pr | [50,51,54] |
| | *Trichodes ornatus tenuosus* Foster | | Megachilidae Sphecidae | | | | [34,51,55] |
| | *Trichodes peninsularis horni* Wolcott & Chapin | | Megachilidae | | | Pr | [39,51] |

*(Continued)*

**Table 1.** (Continued)

| Clerid subfamily and tribe | Species | From galls induced by insects | From nests/cells of solitary bees or wasps | From nests/cells of social Hymenoptera | Other insect prey | Larval feeding mode | References |
|---|---|---|---|---|---|---|---|
| | *Trichodes simulator* Horn | | Apidae Megachilidae | | | Pr, Pa? | [51] |
| Clerinae: Hydnocerini | *Neohydnocera longicollis* (Ziegler) | Tephritidae[Di] Tenthredinidae-[Hy]Gelechiidae[Le] | | | Gall inquilines | Pr, Pa? | [38,45,56,57] |
| | *Neohydnocera tabida* (LeConte) | Tephritidae[Di] | | | Gall inquilines | Pr | [58] |
| | *Phyllobaenus atriplexus* Foster | Cecidomyiidae[Di] | | | Gall inquilines Chalcididae[Hy] Eulophidae[Hy] Eupelmidae[Hy] Eurytomidae[Hy] Platygastridae[Hy] Pteromalidae[Hy] Torymidae[Hy] | Pr | [59–61] |
| | *Phyllobaenus caeruleipennis* (Wolcott) | Tephritidae[Di] | | | Gall inquilines | Pr | [62] |
| | *Phyllobaenus dubius* (Wolcott) | | | | Cephidae[Hy] | Pr | [63] |
| | *Phyllobaenus humeralis* (Say) | Tephritidae[Di] | | | Gall inquilines | Pr | [45] |
| | *Phyllobaenus knausii* (Wickham) | Unident. insect | | | Gall inquilines | Pr | [57] |
| | *Phyllobaenus maritimus* Wolcott | | | | Tortricidae[Le] | Pr | [46] |
| | ***Phyllobaenus obscurus* (Gorham)** | | | Formicidae[Hy] | | **Pa** | **This study** |
| | *Phyllobaenus pubescens* (LeConte) | Unident. insect | Megachilidae | | Gall inquilines Braconidae[Hy] Noctuidae[Le] | Pr | [44,57,64–66] |
| | *Phyllobaenus scaber* (LeConte) | Gelechiidae[Le] | | | Gall inquilines Aphididae[He] Braconidae[Hy] Eurytomidae[Hy] Tortricidae[Le] Psocidae[Ps] Unident. spiders | Pr | [67,68] |
| | *Phyllobaenus unifasciatus* (Say) | Cynipidae[Hy] Unident. insect | | | Gall inquilines | Pr | [37,57] |
| | *Phyllobaenus subaeneus* (Spinola) | | | | Tortricidae[Le] | Pr | [46] |
| | *Phyllobaenus unifasciatus* (Say) | Cynipidae[Hy] Unident. insect | | | Gall inquilines | Pr | [37,57] |
| | *Phyllobaenus verticalis* (Say) | Cynipidae[Hy] | | | Gall inquilines | Pr | [37,38,69] |

*: Putative prey attacked by larvae.

**: Questionable information. According to Foster [51], the references that identify *T. apivorus* as a predator in bee and wasp nests "*were inferred apparently from the specific name or from what was known of* Trichodes *biology in general*". However, there are no reliable reports of bees or wasps serving as larval hosts for *T. apivorus* and the same seems true for *T. nuttalli* for which the only known larval host is the acridid Orthoptera, *Chloealtis conspersa* (Harris).

Di: Diptera; He: Hemiptera; Hy: Hymenoptera; Le: Lepidoptera; Or: Orthoptera; Pa: Parasitoid; Pa?: Facutative parasitoid; Pr: Predator; Ps: Psocoptera; Unident: unidentified.

## Materials and methods

### The ant hosts

The neotropical ant *Ectatomma ruidum* (Roger) has historically been considered as a widely distributed ant taxon throughout the Neotropics, from Mexico to Brazil [74–77], readily visually recognized. These ants have been subject of intense investigation into colony organization, communication, etc., serving as a species model system for several studies including research on intraspecific cleptobiosis, a very rare behavioral trait [78–80]. Recent studies have revealed that this taxon represents indeed a species complex [77,81–84] including at least 6 putative species, which can be separated based on molecular, chemical and acoustic signatures, and on behavioral traits, but with a considerably conserved morphology which has prevented species description. Two species have a wide distribution, from Mexico to northern Argentina (*E. ruidum* sp. 1 and *E. ruidum* sp. 2), and four other species are apparently restricted to few localities in southern Mexico, along the coast of Oaxaca [83,84] where this study was performed (see below). The most abundant and accessible species in the study area was *E. ruidum* sp. 4, which was observed in at least three sites in the municipality of Santa María Tonameca. The species *E. ruidum* sp. 2 (although the most common in Mexico) was only found in a low coastal forest area in the Municipality of Santa María Huatulco, and the distribution of *E. ruidum* sp. 3 was extremely limited, as it is known to occur at only one site (Rancheria Yerba Santa, in the Municipality of San Pedro Mixtepec) that is no longer accessible because it has become private property. For more details on collection sites, see [83,84].

These ants occur in a wide range of habitats in the Neotropics, including plantations and conserved forests, from sea level up to 1600 m [85,86]. Colonies of these species are monodomous with one single exception [87]. Typically, *E. ruidum* nests have a single entrance about 3–4 mm wide dug in the ground [88,89]. In Chiapas, southern Mexico, colonies are made up of 50–200 individuals [90] and colony density can be very high locally (up to 11,200 nests per hectare [91]). These ants have a very generalist diet, and their foraging activity is mainly diurnal [89]. In general, colonies are monogynous, but in some regions they are facultatively polygynous with size-dimorphic queens (macro and microgynes) [92,93].

### Study site, ant sampling, and parasitoid identification

Beetle larvae were collected along with the ants as part of a larger project aimed at characterizing/unveiling cryptic species in the *E. ruidum* complex and assessing the diversity of some of their already recorded parasitoids (parasitic Hymenoptera, Diptera, and Nematodes, see [94]). Collecting an equal number of colonies from the different cryptic species of this ant complex was not a prerequisite for this program, and therefore no effort was made in this regard. Furthermore, the genetic identification of the species could only be carried out retrospectively. Colonies or part of colonies were collected in several sites during three collecting campaigns (November 2015, October 2016 and October/November 2017) in the Pacific Coastal Plain of Oaxaca, Mexico (Table 2). Sampling was carried out in relatively undisturbed areas (very

**Table 2. Summary of the collection sites in the Oaxaca coast (Mexico) and number of colonies of the different *Ectatomma ruidum* species sampled during 2015-2017.**

| *Ectatomma ruidum* species *sensu* Meza-Lázaro et al. [82] | Site | GPS coordinates | Altitud (m asl) | Nb. of sampled colonies (colonies with cocoons) | | |
|---|---|---|---|---|---|---|
| | | | | 2015 | 2016 | 2017 |
| *E. ruidum* sp. 2 | Bajos de Coyula | 15°43′58.8" N, 96°17′24" W | 125 | 5 (4) | 28 (14) | – |
| *E. ruidum* sp. 3 | Yerba Santa | 15°56′28"N, 97°04′27" W | 156 | 4 (4) | 24 (18) | – |
| *E. ruidum* sp. 4 | Piedras Negras | 15°43′49.7"N, 96°40′21.7" W | 18 | – | 22 (13) | – |
| *E. ruidum* sp. 4 | El Zapotal (Mazunte) | 15°41′7.4" N, 96°33′42.5" W | 42 | 4 (4) | 22(9) | 15 (15) |
| *E. ruidum* sp. 4 | Puente Cuatode | 15°43′55.2" N, 96°30′32.4" W | 89 | – | 22 (17) | – |
| | | | | 13 (12) | 118 (71) | 15 (15) |

low urbanization and non-intensive farming) and sites were selected based on the presence of the target ants and their accessibility (both physically and in terms of landowner permission, see "Collection permits" below). Climate is typical of the Pacific watershed lowlands in southern Mexico, i.e., warm sub-humid with precipitation concentrated in the summer months (Aw0 [95]); mean annual temperature is 22–28 °C; mean annual precipitation is ca. 900 mm. The dominant plant communities in the lowlands (50–450 m asl) are classified as seasonally dry tropical forest [96].

In 2015, three people participated in nest excavations, and 13 colonies from three species (E. ruidum sp.1, sp. 2, and sp. 3) were collected from three sites, 12 of them with cocoons. In 2016, six people were available and we were able to sample more sites (five) and excavate more nests (118), with a total of 71 complete colonies collected from the same three species. In 2017, only two people were available. Furthermore, access to certain collection sites was either impossible due to the insecurity prevailing in the region at that time or denied by the owners of the land concerned due to socio-political and socio-economic tensions in the study area, reducing the number of collection sites to just one, where only 15 complete colonies of E. ruidum sp. 4 could be excavated.

At each site, the entrances of ant nests were detected using cookie crumbs as bait and following foraging ants back to their nest. Colonies were excavated and all the individuals present in the nest chambers were collected in plastic bags and transported to the field lab for initial revision. In 2015 and 2017, cocoons of each colony were kept in Petri dishes for a couple of weeks to wait for parasitoid emergence and later inspected and dissected under a stereomicroscope. During the 2016 campaign, when sampling efforts were intensified, cocoons of each colony were separately preserved in 96° alcohol and revised and counted later under a stereomicroscope, as well as the rest of material. Ant larvae were thoroughly examined for the presence of any planidium or evidence of parasitoid attack (presence of scars evidencing a previous unsuccessful attack or signs of endoparasite presence, see [9,11]). All the cocoons collected were externally revised one by one and then were carefully dissected and their contents examined. The identity, number and developmental stage of any parasitoid were recorded. Adult ants were also closely examined for the presence of potential ecto- or endoparasites (e.g., phorid flies, strepsipterans, mites, nematodes, or planidia) attached to their body.

Eucharitid wasps were identified using available keys [97]. Beetles were morphologically identified by one of us (JML). Beetle larvae and the adult reared from cocoons (see Results section) were initially assigned to the same species based on morphology and context. Later, the sequence of the standard barcoding fragment (cytochrome oxidase I, COI) of one larva was obtained. Due to the reduced number of specimens and poor state of the adult, only one larva was extracted. DNA extraction, amplification, and sequencing followed standard protocols [98], using the zooplankton primers developed by Prosser et al. [99]. The COI fragment was sequenced bidirectionally; the sequences were edited using CodonCode v. 3.0.1 (CodonCode Corporation, Dedham, MA, USA). The quality of the forward and reverse reads was assessed using FASTQC [100], with sequence sites evaluated using a minimum quality score threshold of 40. Translation of the 657-nucleotide sequence into a protein sequence produced a chain of 219 amino acids, with no internal stop codons detected. The sequence, along with its corresponding amino acid sequence, was deposited in the Barcode of Life Database (BOLD, boldsystems.org, BOLD:AHE8421) and in GenBank (accession number PX893661). The genetic distances were computed using MEGA 7.0 [101] with the Kimura-2-parameter model.

Voucher specimens of ants (E. ruidum sp. 4: five workers, catalog # F-02128 to F-02132; E. ruidum sp. 3: five workers, catalog # F-02123 to F-02127, and three workers in the same sample vial, catalog # F-03222), the adult beetle and the extracted larva (catalog # AR-0742) were deposited in the Formicidae and Arthropoda collections of El Colegio de la Frontera Sur-Chetumal (ECO-CH-F and ECO-CH-AR, respectively).

## Collection permits

Field sampling complied with the current laws of Mexico (collection permit FAUT-0277 and FAUT-0208 from Secretaría del Medio Ambiente y Recursos Naturales – Dirección General de Vida Silvestre granted to GP-L and to Alejandro Zaldívar-Riverón (UNAM, Mexico) who helped during colony sampling, respectively). No other governmental permit was

required to access the study sites, but formal permission was granted by the owners of the fields where ant nests were excavated.

## Results

### Colony composition and within-nest associated organisms

A total of 146 colonies or part of colonies of three species of the *Ectatomma ruidum* complex (*E. ruidum* sp. 2, sp. 3, and sp. 4 *sensu* Meza-Lázaro et al. [82]) were sampled. Cocoons were present in a total of 98 excavated colonies or part of colonies (18 colonies for *E. ruidum* sp. 2, 22 for *E. ruidum* sp. 3, and 58 for *E. ruidum* sp. 4), and larvae were found in 79 of these colonies (Tables 2, 3, S1). Overall, 11060 adults, 5795 cocoons and 2185 larvae were examined.

A total of 36 nematodes were collected from mermithized workers: three individuals in three different colonies of *E. ruidum* sp. 2, 33 individuals in 11 different colonies of *E. ruidum* sp. 3, and none in *E. ruidum* sp. 4. No larva was parasitized for any of the three species, whilst at dissection eight cocoons were found parasitized by eucharitid wasps belonging to an undetermined species from the *Kapala* Cameron clade. Nine individuals of *Kapala* sp. were obtained: a male pupa from a cocoon of *E. ruidum* sp. 2; one fed planidium and one mature larva from two cocoons from a colony of *E. ruidum* sp. 3; one male pupa, two adult males (on the same host), two adult females, and one adult in poor condition whose sex could not be determined from five cocoons from four colonies of *E. ruidum* sp. 4 (see S1 Table).

In addition, five cocoons were parasitized by characteristic campodeiform beetle larvae (Table 3). In 2015, from a total of 516 cocoons maintained in observation, a single, pink-colored larva emerged 10 days later from isolated cocoons of a colony of *E. ruidum* sp. 3 collected at Ranchería Yerba Santa (representing 0.6% of cocoons of this species for that year, Fig 1A, S1 Video). The cocoon from which the larva had emerged showed signs of a single round exit hole. When it was dissected, the ant host pupa was found beheaded and only its empty exoskeleton remained (Fig 1B). The larva was isolated in a glass vial along with some fabric to help pupation and an adult belonging to a species of Cleridae (Insecta: Coleoptera) was reared. No other beetle specimens were obtained from the other cocoons after dissection. During the 2016 campaign, out of a total of 4,208 cocoons, four cocoons from *E. ruidum* sp. 4 collected at Puente Cuatode were parasitized by beetle larvae or presented signs of attack (Table 3). One second- and two third-instar larvae were obtained (Figs 2A–2C), each one found inside an intact cocoon on an ant pupa. Another cocoon presented a round hole, suggesting a larva had already left the host to pupate elsewhere and the remains of the host pupa were found inside (Fig 2D arrow). During 2017 only one site was sampled (El Zapotal, Mazunte) and 15 colonies of *E. ruidum* sp. 4 with 1071 cocoons were collected. No beetle or parasitoid wasp was found during this collecting campaign. Whatever the campaign, no eggs or triungulin larvae of Cleridae have ever been observed.

**Table 3. Infestation rate of *Ectatomma ruidum* spp. cocoons by *Phyllobaenus obscurus*.**

| Year of collection | Species | Locality | Nb. of colonies with cocoons | Total nb. of cocoons | Total nb. of larvae | % of cocoon infestation (exact proportion) |
|---|---|---|---|---|---|---|
| 2015 | *E. ruidum* sp. 2 | Bajos de Coyula | 4 | 111 | 2 | 0 |
| | *E. ruidum* sp. 3 | Yerba Santa | 4 | 172 | 72 | 0.6 (1/172) |
| | *E. ruidum* sp. 4 | El Zapotal | 4 | 233 | 87 | 0 |
| 2016 | *E. ruidum* sp. 2 | Bajos de Coyula | 14 | 487 | 74 | 0 |
| | *E. ruidum* sp. 3 | Yerba Santa | 18 | 1285 | 794 | 0 |
| | *E. ruidum* sp. 4 | Puente Cuatode | 17 | 1229 | 371 | 0.3 (4/1229) |
| | | El Zapotal | 9 | 717 | 116 | 0 |
| | | Piedras Negras | 13 | 490 | 191 | 0 |
| 2017 | *E. ruidum* sp. 4 | El Zapotal | 15 | 1071 | 478 | 0 |
| Total | | | 98 | 5795 | 2185 | < 0.1 (5/5795) |

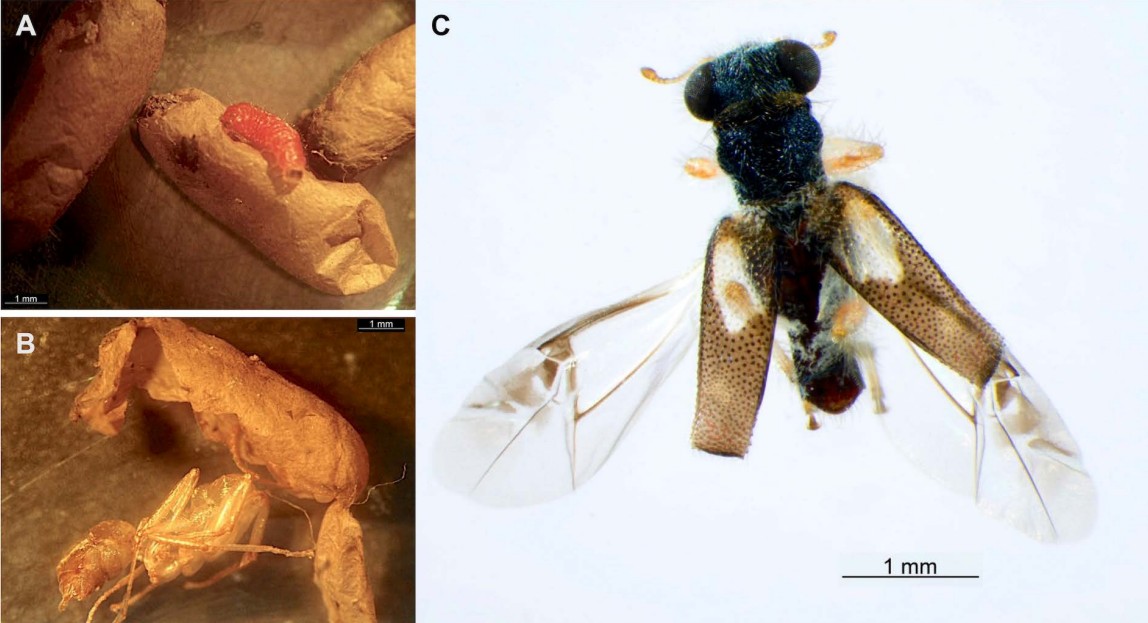

**Fig 1. *Phyllobaenus obscurus* (Coleoptera: Cleridae). (A)** a third-instar larva that has just emerged from the host cocoon; **(B)** remains of the *E. ruidum* sp. 3 pupa from which the mature beetle larva has emerged (the dissected cocoon can be observed); **(C)** *Phyllobaenus obscurus* adult reared from the larva that parasitized the pupa in (B). Photo credits: (A-B) Gabriela Pérez-Lachaud, (C) Humberto Bahena-Basave.

## Beetle parasitoid identity

Based upon morphology, the adult could be identified as a specimen of *Phyllobaenus obscurus* (Gorham) (Cleridae: Hydnocerinae: Hydnocerini) (Fig 1C). The COI fragment supports assignment of the larva to the Cleridae, and is consistent with the genus *Phyllobaenus* (Fig 3). Blasting the sequence of our larva on the Genbank site returned the sequence of *Phyllobaenus* cf. *bituberculatus* (NLG-2013 voucher 4694, Genbank accession number KC524584.1) as the closest available sequence with 88.74% of identity (best score of 804). The genetic distance based on Kimura-2-parameter model between these two sequences is $0.116 \pm 0.013$. Unfortunately, the poor state of the beetle adult prevented extraction, and no sequence could be obtained.

## Discussion

Here we provide strong evidence that a checkered beetle behaves as parasitoid of the pupae of ants of the *E. ruidum* species complex in Oaxaca, Mexico. The adult was identified as *Phyllobaenus obscurus* based on morphology, while the COI fragment of the larva supports assignment to Cleridae and is consistent with *Phyllobaenus.* There is no public COI sequence for *P. obscurus* available for comparison with our larva and thus species level confirmation awaits a reference sequence for this taxon in the future. However, the evidence presented here supports a reliable connection between the larva and adult beetle based on morphology, rearing, and co-occurrence; so, the most parsimonious hypothesis is that the larvae and adult beetle found in this study belong to the same species of *Phyllobaenus*.

Although the eggs or triungulin larvae of the clerid beetle were not found in association with these ants, our observations showed that the larva of *P. obscurus* developed at the expense of a single ant pupa enclosed in an intact cocoon (neither pierced nor opened), resulting in the death of the host after the clerid larva had finished feeding. These facts do not correspond in any way to predation cases reported for many other beetles, such as rove beetles and other checkered

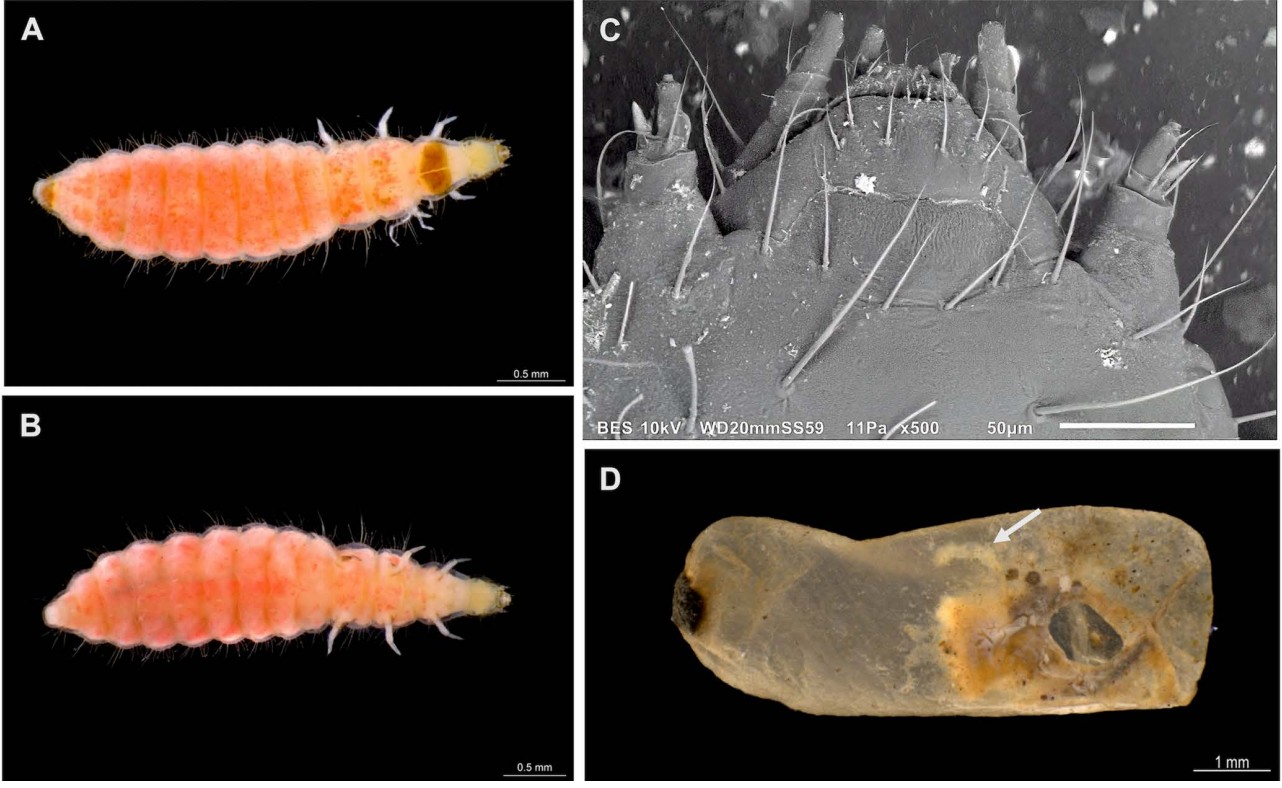

**Fig 2. *Phyllobaenus obscurus* larvae and remains of the host within the host cocoon.** (A) second-instar *P. obscurus* larva, dorsal view; **(B)** same, ventral view; **(C)** dorsal view of the head of a second-instar larva, low vacuum SEM photography; **(D)** an *E. ruidum* sp. 4 cocoon showing an exit hole after the departure of a *P. obscurus* larva (the ant pupa remains are visible, the white arrow pointing at a leg). Photo credits: (A,B,D) Humberto Bahena-Basave; (C) Gabriela Pérez-Lachaud and Manuel Elías-Gutiérrez.

beetles, where several hosts are preyed upon before development is completed. In our observations, within each cocoon, only one host was available and large enough to allow the development of a parasitoid the size of *P. obscurus*. The genus *Phyllobaenus*, the second largest Hydnocerini genus after *Callimerus* Gorham, comprises 114 described species [102–105]. Hydnocerini are considered general ant mimics, due either to their wasp-waisting color patterns, their elongate body shape (also sometimes creating a physical "waisting" effect), or their very movement patterns that are similar to those of an ant running quickly up and down plant substrates and changing directions erratically [32,106–108]. For example, *Neohydnocera aegra* (Newman) strikingly resembles a species of *Pseudomyrmex* Lund and has been found abundantly among these ants [109], and *Phyllobaenus cryptocerinus* (Gorham) is a reported mimic of *Cephalotes* Latreille ants (= *Cryptocerus* Latreille) [106], but no species from this clade has ever been reported as an intranidal symbiont of ants. Our findings constitute the first record of a beetle species adapted to parasitize ant brood within the defended boundaries of the nest of rather aggressive ants.

Like many of its congeners, *P. obscurus* is an active clerid with a highly variable color pattern and body size (2–5 mm). This species is known from Mexico, Guatemala, Costa Rica, and Panama [110–113]. Araujo et al. [114] included Brazil based on an unvouchered record with no data nor depository presented from a personal database (see "Unverified Distributions" in [115]). Prior to this study it was known only from Durango, Morelos and Tamaulipas in Mexico [116,117]. The data presented herein for Oaxaca and an unpublished record from Chiapas ([Verbatim label data] 3000', "El Chorreadero" by Hwy. 190, 1.25 km E Tuxtla Gutierrez, under debris, gravel bank of stream in deep ravine, 25-XII-1972, H. Frania,

PLOS One | https://doi.org/10.1371/journal.pone.0335300   March 20, 2026

10 / 19

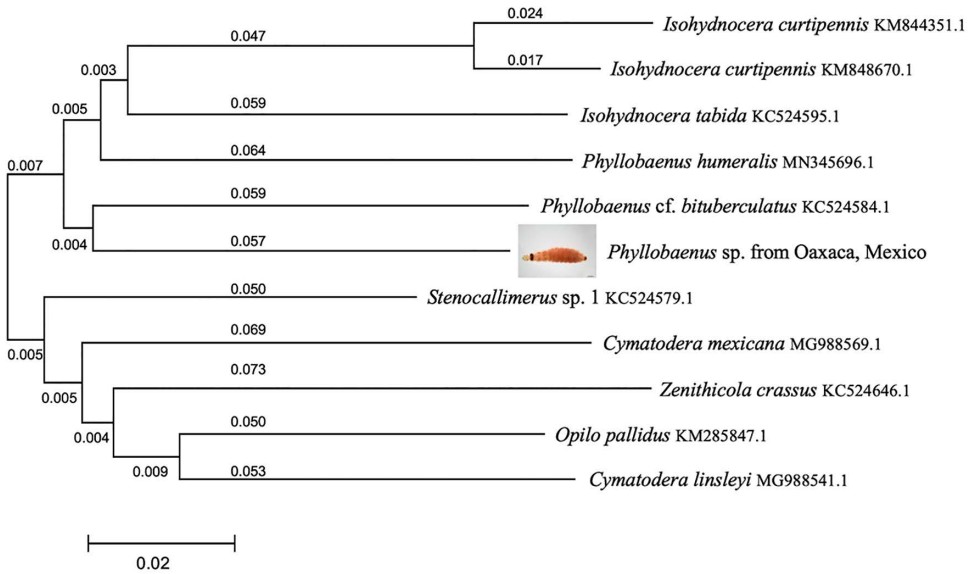

**Fig 3. Illustrative neighbor-joining tree diagram showing the genetic relationship between our sample and the 10 most identical taxa from Blast, based on Kimura-2 genetic distances.** Numbers indicate branch lengths. Species names are followed by Genbank accession numbers and correspond to those in Genbank platform; however, some species have been synonymized as follows: *Stenocallimerus* is a synonym of *Callimerus*, *Isohydnocera tabida* is a synonym of *Neohydnocera tabida*, and *Isohydnocera curtipennis* is a synonym of *Neohydnocera longicollis*. The scale refers to Kimura 2P distance. Photo credits: Humberto Bahena-Basave.

FSCA, Florida State Collection of Arthropods) are new state records for Mexico. Despite the broad distribution of this rather common species, nothing was known about its natural history prior to our findings, and the larva and larval feeding habits were unknown.

Most specimens of *P. obscurus* have been collected by beating the foliage of live trees and occasionally shrubs (JML personal observations; accounts from numerous beetle collectors). Specimen label data indicate few specimens collected with other methods (e.g., lights, fogging, Malaise trap) (Leavengood, unpublished data). Notably, label data for the Chiapas new state record for Mexico (see above) indicate an adult was caught on the ground and under debris, which is most unusual. In the Western Hemisphere, hydnocerines are not observed on the ground, but usually high in foliage (e.g., [108]) and there are no records of *Phyllobaenus* collected in pitfall traps. Author JML has been presented two separate accounts of a *Phyllobaenus* specimen emerging from an ant hill in the ground. But no specimens were produced nor identifications made. So, with only one vouchered record (Chiapas, FSCA), two unvouchered events (ant hill emergence observations), and the five brood parasitoid observations noted herein (recorded in different years, at two different sites 65 km apart, and concerning two closely related but nevertheless different ant species), any record of *Phyllobaenus* at ground level (or below) is highly unusual, yet all but one such record are specifically associated with ants. Furthermore, it should be noted that our records concern clerid larvae that were sampled inside ant nests, in deep chambers, and developed on ant pupae enclosed in intact silken cocoons, which excludes the possibility of any accidental or fortuitous presence of the larvae within the ant nests.

Myrmecophilous taxa are not scattered randomly across the arthropod tree of life. Rather, their evolution is strongly biased to certain groups and, among insects, towards the Coleoptera which "*contains a large pool of lineages with the potential for transitioning to myrmecophily*" [27]. At least 33 of the beetle families, representing approximately one-fifth, have thus far been documented to include one or more taxa that exhibits close association with ant colonies [27]. Many of the most familiar beetle families contain at least one lineage that inhabits ant colonies, including the mainly aquatic beetle

family Hydrophilidae [118]. Indeed, at least two species of *Sphaerocetum* Fikáček have been collected in mixed nests of *Camponotus* Mayr and *Crematogaster* Lund, and a larva of *S. arboreum* Fikáček et al. was collected in an ant nest along with the adults, implying the entire life cycle may take place inside the ant nest, suggesting that these beetles likely prey on ant larvae [118].

Ants and their nests are reliable, fixed resources in space and time [119]. Furthermore, ant brood in tropical species is present all year round so it is not surprising that a diverse array of parasitoids has adapted to ants as hosts, and new family-level records of species parasitizing ants or their brood are recorded from time to time (e.g., [10,120–122]). In addition, as suggested by Parker [27], several ecological constraints and phenotypic attributes may predispose some taxa to undergo the evolutionary transition from free-living to myrmecophily which, in combination, could explain the myrmecophile beetle bias: i) the huge species richness of Coleoptera, ii) their ecological predisposition to encountering and exploring ant colonies, and iii) their possession of a major protective preadaptation in the form of elytra. Additionally, the size of the myrmecophile relative to the ant host species (small myrmecophile species/large host), a limuloid body form, and other protective morpho-anatomical adaptations (i.e., pedofossae) are traits that facilitate the integration of some intruders into colony life (see [13,123–125]).

Although we clearly established that the larvae of this clerid develop as parasitoids, whether this species is a facultative or obligate parasitoid of the ant brood remains to be clarified. The line between predation, parasitoidism, and parasitism is often quite blurred, and some species may even switch between strategies depending on the context. For example, the mite *Macrodynichus sellnicki* (Hirschmann & Zirngiebl-Nicol) is a common parasitoid of the pupae of the invasive ant *Nylanderia fulva* (Mayr), but can also develop on a larger host, the native *E. ruidum* sp. 2 in Colombia [15]. While solitary attacks do not result in the death of the host, mites attacking in groups drain the host of its internal fluids and kill it. Therefore, the nature of the parasitic or parasitoidic relationship between *M. sellnicki* and *E. ruidum* sp. 2, is context dependent, varying according to the intensity of the attack. Similarly, the larvae of the chechered beetle *T. umbellatarum*, approaches the boundary between predation and parasitoidism [70,71], demonstrating here too a context-dependent change in lifestyle. Elucidating the lifestyle of *P. obscurus*, whether a facultative or obligate parasitoid, warrants further investigation.

Even though the prevalence of parasitism in certain ant parasitoids can sometimes reach high levels (e.g., [125–128]), extremely low prevalence levels, such as those observed in this study, are quite common among many highly specialized parasites and parasitoids of social insects. For example, only eight cocoons were found parasitized by *Kapala* eucharitid wasps in this study and various examples of extremely low prevalence of parasitism have previously been reported for many other ant associates [9,129–135]. Plausible explanations for rarity include low host encounter rate, strong host colony defenses, seasonality, or sampling detection limits, for example. In addition, given the broad geographic range of *P. obscurus* compared to the restricted distribution of *E. ruidum* sp. 3 and sp. 4, further research is needed to address the possibility of specialized local host use if the beetle uses different hosts across regions or if suitable conditions are patchy. Coevolution between ants and their rare, specialized parasitoids is intriguing because, by definition, a parasitoid kills its host. However, lethal outcomes for individual host ants rarely result in the destruction of the colony, and parasitoidism in ants resembles more to a chronic disease, allowing for stable coexistence between parasitoids and ants. This is certainly one of the reasons why so few parasites and parasitoids actually have the potential to act as biological control agents against pest ants.

Our finding of an unusual feeding mode for a checkered beetle is surprising but not entirely unexpected as other checkered beetles belonging to different tribes border the line between predator and parasitoid [42,51,52,57,70,71], likely depending on the relative size of their prey/host. It suggests that the parasitoid lifestyle may have evolved several times within the Cleridae. Notably, our record is also the first evidence of beetles parasitizing successfully the immature stages of ants within their subterranean nests, suggesting a major evolutionary shift in the biology of Hydnocerini. Our results further support a previously stated hypothesis about the evolution of ant parasitoidism in clades with predatory habits. As suggested for other unusual ant parasitoids such as the microdontine syrphid fly *Hypselosyrphus trigonus* Hull [121], the

only parasitoid species among this clade of predators, it is likely that parasitoidism in *P. obscurus* evolved from a predatory ancestor. According to Godfray [136], the transition from a predatory to a parasitoid lifestyle may be straightforward since a predator that requires only a single prey item for its development is, by definition, a parasitoid.

There are still some gaps in knowledge regarding the biology of this clerid. Like other ant parasitoids such as eucharitid wasps of the genera *Kapala*, *Isomerala* Shipp, *Obeza* Heraty, or *Pseudochalcura* Ashmead [8], *P. obscurus* develops at the expense of host larvae/pupae under the protection of a cocoon, avoiding deadly interactions with their host ants. But, as our observations show, when larval development is completed, the larva exits the host cocoon to pupate. Clerids are known to commonly pupate near their prey (e.g., in galleries of their prey, or tree crevices nearby or inside the host pupal cell [57,58]), but the site of pupation of *P. obscurus* remains to be uncovered. Its high mobility and small size (only 2–5 mm for the adult) relative to the *Ectatomma* hosts (8–9 mm) may help avoiding ant aggressiveness. Infiltration into the ant nest is probably by phoresis of the mobile first-instar larva (triungulin) as occurs in other ant parasitoids which lay eggs away from the host [7,20,22], but this needs confirmation.

Although some checkered beetle species are considered of importance as potential biological control agents, there are very few rearing records for Cleridae in general, and data on their natural life history are scarce (Table 1). However, results of laboratory and field studies underscore the plasticity and adaptability in the food habits of Cleridae (e.g., [54] and references in Table 1). The shift in foraging behavior reported here, from predation towards parasitoidism, is likely to confirm the diversified feeding traits within the Cleridae.

## Supporting information

**S1 Video. A *Phyllobaenus obscurus* larva that has just emerged from its *Ectatomma ruidum* sp. 3 host cocoon.** (MOV)

**S1 Table. Detailed collection data for the three species of the *Ectatomma ruidum* complex studied: species concerned, location, date of collection, colony composition.** µQ: number of microgynes. (XLSX)

## Acknowledgments

We are grateful to Ruby Meza-Lázaro, Alejandro Zaldívar-Riverón (UNAM, Mexico) and Carlos Santamaría (Universidad del Valle, Cali, Colombia) for their assistance with field work; to Humberto Bahena-Basave, Holger Weissenberger, Manuel Elías-Gutiérrez, and Alma Estrella García-Morales (Ecosur) for their help with insect pictures, illustration in a previous version, SEM photography, and DNA extraction, amplification and sequence alignment, respectively. We thank Roland Gerstmeier (Zoologische Staatssammlung München, Germany) for his updated estimate of the number of clerid species. We greatly appreciate the thoughtful and detailed comments provided by five reviewers and the academic editor which helped improve the clarity, accuracy, and overall quality of the article.

## Author contributions

**Conceptualization:** Gabriela Pérez-Lachaud, Chantal Poteaux, Jean-Paul Lachaud.

**Data curation:** Gabriela Pérez-Lachaud, Chantal Poteaux, John M. Leavengood Jr., Jean-Paul Lachaud.

**Formal analysis:** Gabriela Pérez-Lachaud, Jean-Paul Lachaud.

**Funding acquisition:** Chantal Poteaux, Jean-Paul Lachaud.

**Investigation:** Gabriela Pérez-Lachaud, Chantal Poteaux, John M. Leavengood Jr., Jean-Paul Lachaud.

**Methodology:** Gabriela Pérez-Lachaud, Jean-Paul Lachaud.

**Project administration:** Gabriela Pérez-Lachaud.

**Supervision:** Gabriela Pérez-Lachaud, Jean-Paul Lachaud.

**Validation:** Chantal Poteaux, John M. Leavengood Jr..

**Writing – original draft:** Gabriela Pérez-Lachaud, Jean-Paul Lachaud.

**Writing – review & editing:** Gabriela Pérez-Lachaud, Chantal Poteaux, John M. Leavengood Jr., Jean-Paul Lachaud.

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
