## [Decision Letter · Decision Letter 0]

23 Nov 2025

Dear Dr. Lachaud,

Thank you for submitting your manuscript to PLOS ONE. After careful consideration, we feel that it has merit but does not fully meet PLOS ONE’s publication criteria as it currently stands. Therefore, we invite you to submit a revised version of the manuscript that addresses the points raised during the review process.

We look forward to receiving your revised manuscript.

Kind regards,

Petr Heneberg

Academic Editor

PLOS ONE

Journal Requirements:

This study was partially funded by the project number M12A01, Ecos-Nord-CONACYT Program granted to CP and J-PL.

This study was partially funded by the project number M12A01, Ecos-Nord-CONACYT Program granted to CP and J-PL.

5. We note that Figure 1 in your submission contain map images which may be copyrighted. All PLOS content is published under the Creative Commons Attribution License (CC BY 4.0), which means that the manuscript, images, and Supporting Information files will be freely available online, and any third party is permitted to access, download, copy, distribute, and use these materials in any way, even commercially, with proper attribution. For these reasons, we cannot publish previously copyrighted maps or satellite images created using proprietary data, such as Google software (Google Maps, Street View, and Earth). For more information, see our copyright guidelines: http://journals.plos.org/plosone/s/licenses-and-copyright.

Additional Editor Comments:

The claim that this is the first beetle species acting as a parasitoid of ants may be too definitive. Several beetle groups show predatory or brood-infiltrating behaviors inside ant nests, and some approach the boundary between predation and parasitism. Without a clear explanation of how those cases are excluded, the statement risks sounding broader than the available evidence supports.

The line separating predation, facultative parasitoid behavior, and true parasitoid development is not sharply drawn in the manuscript. Cleridae are known for flexible feeding strategies, and the observations described could fit multiple life history modes. The interpretation would benefit from clearer criteria distinguishing these possible strategies to support the argument for parasitoidism.

The absence of triungulin larvae, eggs, or other early developmental stages associated with ant nests leaves an important gap. Many Cleridae with parasitic or phoretic life cycles rely on triungulin stages for host access, so not finding these forms makes the developmental pathway uncertain. Further clarification of early stages would help support the proposed life history model.

The identification of larvae as belonging to the same species as the reared adult remains tentative without molecular confirmation. Only one adult was successfully obtained, and the remaining larvae are attributed to the same species based solely on morphology and context. Genetic verification would help solidify this assignment and reduce uncertainty regarding species identity.

Some sections present evolutionary explanations in a way that can appear more conclusive than the evidence permits. Arguments about predispositions toward myrmecophily or transitions toward parasitoidism would benefit from phrasing that distinguishes speculation from established fact. Clarifying these boundaries would provide a more balanced interpretation.

The discussion of how larvae or adults avoid ant aggression is presented without accompanying behavioral observations. Without direct evidence of interactions between the beetle life stages and the ants, these explanations rely on inference rather than documentation. A more cautious description would better reflect what is known and what remains to be studied.

Reviewers' comments:

Reviewer's Responses to Questions

**Comments to the Author**

1. Is the manuscript technically sound, and do the data support the conclusions?

Reviewer #1: Yes

Reviewer #2: Yes

Reviewer #3: Yes

Reviewer #4: Yes

Reviewer #5: Partly

2. Has the statistical analysis been performed appropriately and rigorously?

Reviewer #1: N/A

Reviewer #2: N/A

Reviewer #3: I Don't Know

Reviewer #4: N/A

Reviewer #5: N/A

3. Have the authors made all data underlying the findings in their manuscript fully available?

Reviewer #1: Yes

Reviewer #2: Yes

Reviewer #3: Yes

Reviewer #4: Yes

Reviewer #5: Yes

4. Is the manuscript presented in an intelligible fashion and written in standard English?

Reviewer #1: Yes

Reviewer #2: Yes

Reviewer #3: Yes

Reviewer #4: Yes

Reviewer #5: Yes

Reviewer #1: Thank you for inviting me to review this interesting and well-documented article.

The authors reported a rare case of parasitism from the predatory Phyllobaenus obscurus beetles on the Ectatomma ruidum species of ant cocoons. The discovery represents a novel adaptation for the beetle family.

The presentation of the complex and yet-to-be-further-classified ant host species adds more interest to the study findings. Based on molecular, chemical, and acoustic monikers, and on behavior, the existence of cryptic species is strongly suggested—an ant species whose description is difficult due to the high similarity of morphometric parameters. If the main objective is to demonstrate the parasitism trait, it gives an insight into further opportunities for taxonomic investigation of the species known for food thievery (cleptobiosis). This kind of research is valuable to the academic and student researcher community as a whole. For this reason, I strongly recommend the study for consideration for publication.

However, the complex nature of the topic requires robust knowledge to understand it at best, hence I provided some basic suggestions to improve and facilitate the first-time readers' comprehension.

See the attached review document please.

Reviewer #2: An interesting paper that should be published. In the absence of actually seeing beetle larvae attack and develop on the ants, the presented information is as good evidence of parasitoidism in this system as could be expected. This paper is well written, and I cannot find anything significant I would want to see changed.

My only comment is italicize Trichodes on L 487.

Reviewer #3: The article has been written very well first time reporting the parasitization of Ectatomma ants by the beetle Phyllobaenus obscurus. This novel information will enrich the knowledge of Myrmecophily evolution in Checkered beetle belonging to the family Cleridae. I have found one typographical mistake which has been marked in the reviewed original manuscript attached with this report. I suggest you to improve the quality of the photographs in Fig 2. The article deserves publication after minor correction. I congratulate all the authors for writing this article.

Reviewer #4: Comments to the Author,

The manuscript "Ant parasitoidism in checkered beetles: Phyllobaenus obscurus developing inside intact cocoons of two species of the Ectatomma ruidum species complex" deals with the new record of Phyllobaenus obscurus as a parasitoid of ants of the Ectatomma ruidum species complex. The manuscript is generally very well written and easy to read. The comments I have to improve the manuscript and its presentation are, in my opinion, easy to address. The most significant would relate to the details of the methodology and the results. For example, it is not mentioned how many colonies were collected for recording until the emergence of the adult parasitoids, nor how many colonies were separated for dissection (details in the PDF). Regarding methodology and results, several aspects lack detail (see details in the PDF), for example, why are there many more colony records in 2016 compared to 2015? Was there a greater effort in the collections? Is that due to a larger area covered or a greater number of people recording colonies?

Finally, I think the discussion is quite long and doesn't always address the actual field results. Furthermore, issues related to the low rate of parasitism weren't discussed or even compared with other studies (details of which are also included in the PDF).

Reviewer #5: The paper deals with a potential case of ant parasitoidism by a clerid beetle, this is quite interesting since if confirmed would be the first case among coleoptera. The text is clear, and a good literature overview has been reported, although I suggest the Authors to better develop the case of other myrmecophilous parasite beetles (e.g. staphylinids) whose larvae feed on ant brood as a comparative case study.

The study is primarily descriptive and reports a general and detailed check of many ant colonies of the putative host species. The sample is quite large but only in a few cases beetle larvae emerging form ant cocoons (suggesting a case of parasitoidism) where recorded. Given the few evidence and the lack of information on the habits of adults, I suggest the Authors to not emphasize excessively the certain discovery of a new species of ant parasitoid. Hence, I suggest to better modulate the discussion, as well as the title of the paper accordingly, limiting to what exactly the observational data allow to state, such as first data on a possible case of parasitoidism by coleoptera on ant brood.

A couple of other minor points on the text:

Lines 69-70: Diplopoda is not an Order.

Lines 138-140: revise the writing, the sentence is not clear.

**Do you want your identity to be public for this peer review?** For information about this choice, including consent withdrawal, please see our Privacy Policy

Reviewer #1: **Yes:** EXELIS MOISE PIERRE

Reviewer #2: No

Reviewer #3: **Yes:** Dr. Suprakash Pal

Reviewer #4: No

Reviewer #5: No

---

## [Author Response · Author response to Decision Letter 1]

26 Jan 2026

[All references to lines in the revised version text refer to the revised version with tracked changes]

Journal Requirements

R: This has been checked.

R: Done (see l. 263-272 in the revised version).

This study was partially funded by the project number M12A01, Ecos-Nord-CONACYT Program granted to CP and J-PL.

R: Done (see the Cover Letter).

This study was partially funded by the project number M12A01, Ecos-Nord-CONACYT Program granted to CP and J-PL.

R: Done (see the Cover Letter).

5. We note that Figure 1 in your submission contain map images which may be copyrighted. All PLOS content is published under the Creative Commons Attribution License (CC BY 4.0), which means that the manuscript, images, and Supporting Information files will be freely available online, and any third party is permitted to access, download, copy, distribute, and use these materials in any way, even commercially, with proper attribution. For these reasons, we cannot publish previously copyrighted maps or satellite images created using proprietary data, such as Google software (Google Maps, Street View, and Earth). For more information, see our copyright guidelines: http://journals.plos.org/plosone/s/licenses-and-copyright.

R: All permission letters have been joined (see uploaded files noted as "Other files").

R: All articles suggested by one of the reviewers (Reviewer #1) were checked before responding to his comments.

R: This has been checked.

Additional Editor Comments

The claim that this is the first beetle species acting as a parasitoid of ants may be too definitive. Several beetle groups show predatory or brood-infiltrating behaviors inside ant nests, and some approach the boundary between predation and parasitism. Without a clear explanation of how those cases are excluded, the statement risks sounding broader than the available evidence supports.

The line separating predation, facultative parasitoid behavior, and true parasitoid development is not sharply drawn in the manuscript. Cleridae are known for flexible feeding strategies, and the observations described could fit multiple life history modes. The interpretation would benefit from clearer criteria distinguishing these possible strategies to support the argument for parasitoidism.

R: While parasitoidism was clearly defined in the previous version (lines 44-46), its distinction from predation was not clearly stated. We have corrected this omission (l. 52-53 of the revised version). We also agree that the discussion in the previous version of our manuscript did not sufficiently emphasize the arguments in favor of interpreting our results as evidence of parasitoidism by P. obscurus, and we thank you for pointing this out. We have thoroughly revised the discussion, rewording some parts and, we hope, presenting strong arguments in favor of this lifestyle in a more convincing manner. In addition, we now present further evidence that reduces uncertainty regarding the species identity. The (COI) sequence of one larva was obtained and used to connect the adult and the early stages of this species.

Although the number of cases reported here is small, the evidence currently available clearly indicates that we are dealing with a case of parasitoidism. Indeed, each of the three P. obscurus larvae found on an ant pupa was enclosed in an intact cocoon (neither pierced nor opened) of E. ruidum sp. 4, suggesting that an early larval stage was already anchored to the ant larva when the latter weaved its cocoon. Furthermore, at least one adult was reared from a larva that had emerged from a cocoon of E. ruidum sp. 3 by piercing an exit hole. This larva pupated and attained adulthood without further feeding upon ant larvae or pupae. The fact that P. obscurus eggs or any triungulin larvae have never been found inside E. ruidum nests does not detract from the fact that each beetle larva developed at the expense of a single ant pupa, which as a result was emptied of its substance and died, which is the very definition of the term parasitoid. The facts observed and reported in our manuscript do not correspond in any way to the cases of predation reported for many other beetles, such as rove beetles and other checkered beetles, where several individual hosts are commonly preyed upon. In our case, inside the cocoons, only one host was available and large enough to allow the development of at least one parasitoid the size of P. obscurus. These arguments have been presented and discussed in a paragraph added to the discussion.

The absence of triungulin larvae, eggs, or other early developmental stages associated with ant nests leaves an important gap. Many Cleridae with parasitic or phoretic life cycles rely on triungulin stages for host access, so not finding these forms makes the developmental pathway uncertain. Further clarification of early stages would help support the proposed life history model.

R: We agree with the editor that the most likely mode of infestation of nests is via triungulin larvae, as is the case with various other parasitic Cleridae. We had already included a sentence in the discussion clarifying this point (see l. 334-336 in the previous version).

This study is part of a larger project focusing on genetic/behavioral/acoustic/chemical/morphological variability of species in the Ectatomma ruidum complex. At the time of collection, we were unaware of the presence of the clerid, and the workers and larvae were only inspected externally. The planidia of some eucharitids may be found in the infrabucal pocket of foraging workers and then transmitted to the ant larvae during feeding. We did not check the workers as thoroughly and this will need to be done in the future. However, as noted above, P obscurus larvae were found feeding on ant pupae inside their cocoons. Since the cocoon is woven by the late-instar ant larva, we assume that the triungulin had already attached itself to its host at that moment and was enclosed in the cocoon where it was invisible to the workers of these aggressive ants.

The identification of larvae as belonging to the same species as the reared adult remains tentative without molecular confirmation. Only one adult was successfully obtained, and the remaining larvae are attributed to the same species based solely on morphology and context. Genetic verification would help solidify this assignment and reduce uncertainty regarding species identity.

R: Thank you for highlighting this point. You are right, when we submitted our previous version, we assigned the adult and the larvae to the same species based solely on morphology and context, comforted by our experience and knowledge of ant parasitoids. However, we were in the process of sequencing some specimens. Unfortunately, not all specimens amplified. We have now obtained a sequence for one of the beetle larvae, which is close to, but distinct from, the sequences of several Phyllobaenus species available in public databases. We have added this information both in the “Material and Methods” and “Results” sections (see l. 248-256 and 327-337 of the revised version). Unfortunately, we were unable to obtain a sequence for the adult (poor condition of the voucher and fungal contamination), and there are no sequences for P. obscurus in public databases. We hope that, despite the sequencing problems, the new results will reduce uncertainty regarding the identity of the species.

Some sections present evolutionary explanations in a way that can appear more conclusive than the evidence permits. Arguments about predispositions toward myrmecophily or transitions toward parasitoidism would benefit from phrasing that distinguishes speculation from established fact. Clarifying these boundaries would provide a more balanced interpretation.

The discussion of how larvae or adults avoid ant aggression is presented without accompanying behavioral observations. Without direct evidence of interactions between the beetle life stages and the ants, these explanations rely on inference rather than documentation. A more cautious description would better reflect what is known and what remains to be studied.

R: Thank you for bringing this issue to our attention. We have deleted some sentences and reworded some paragraphs accordingly. We have also expanded on the points that should be clarified in future studies.

Reviewers' comments

Reviewer #1:

Thank you for inviting me to review this interesting and well-documented article. The authors reported a rare case of parasitism from the predatory Phyllobaenus obscurus beetles on the Ectatomma ruidum species of ant cocoons. The discovery represents a novel adaptation for the beetle family.

The presentation of the complex and yet-to-be-further-classified ant host species adds more interest to the study findings. Based on molecular, chemical, and acoustic monikers, and on behavior, the existence of cryptic species is strongly suggested—an ant species whose description is difficult due to the high similarity of morphometric parameters. If the main objective is to demonstrate the parasitism trait, it gives an insight into further opportunities for taxonomic investigation of the species known for food thievery (cleptobiosis). This kind of research is valuable to the academic and student researcher community as a whole. For this reason, I strongly recommend the study for consideration for publication.

However, the complex nature of the topic requires robust knowledge to understand it at best, hence I provided some basic suggestions to improve and facilitate the first-time readers' comprehension.

See the attached review document please.

Additional Comments:

Abstract

Well presented. It helped to understand the subject of research for a novice reader.

With the low occurrence of inferior to 0.6% parasitism of ant brood, could this be a case of accidental disorientation? Or just a facultative parasitism strategy for survival benefit? The authors clearly stated that the facultative or obligate pattern of the beetles is not defined in this study and needs to be carried out in the future. However, the absence of more comments on this poor occurrence value left me pondering on the elusive character of this newly discovered opportunistic parasite. See more comments in the methodology and results section review.

This point could be discussed briefly at the end of the discussion with robust citations to support it.

R: It seems unlikely that we observed larvae inside intact cocoons by chance or that this was a case of accidental disorientation, especially since ants of the genus Ectatomma are efficient opportunistic predators and attack most insects they encounter, and even more so if they are near or inside their nests (see Dejean and Lachaud 1992 [Insect Soc. 1992;39(2):129–143]; Schatz et al. 1996 [Insect Soc. 1996;43(2):111–118]). It is therefore completely implausible that a beetle such as P. obscurus, or any other foreign individual, could land by chance (or

---

## [Editor Report · Decision Letter 1]

4 Feb 2026

Dear Dr. Lachaud,

Thank you for submitting your manuscript to PLOS ONE. After careful consideration, we feel that it has merit but does not fully meet PLOS ONE’s publication criteria as it currently stands. Therefore, we invite you to submit a revised version of the manuscript that addresses the points raised during the review process.

We look forward to receiving your revised manuscript.

Kind regards,

Petr Heneberg

Academic Editor

PLOS One

Journal Requirements:

Additional Editor Comments:

The adult specimen identified morphologically as Phyllobaenus obscurus was not sequenced, because extraction failed due to poor condition, and no adult COI is available for direct comparison. This leaves a critical gap in the identity chain: the adult diagnosis rests on morphology alone, and the larval barcode cannot be anchored to that adult voucher. The statement that DNA data connect adult and immature stages is not supported when the adult sequence is missing. A practical improvement is to revise the text to separate three claims: adult identification by morphology, larval placement by COI, and the inferred association based on rearing and co-occurrence.

The COI BLAST outcome reported in the Results creates a second identity vulnerability. The closest available sequence is described as only 88.74% identical, paired with a Kimura 2-parameter distance of about 0.116, which is far outside typical species-level barcode expectations for insects and can be large even for many congeneric comparisons. That pattern does not automatically invalidate genus placement, but it does mean that phrases like “placed it without any ambiguity” and “connect adult and immature stages” are overstated relative to the reported similarity metrics. A more defensible interpretation is that the COI fragment supports assignment to Cleridae and is consistent with Phyllobaenus, with species-level confirmation pending a reference sequence for P. obscurus and successful sequencing of the adult voucher.

The Discussion repeats a strong linkage claim that is difficult to justify given the single larval sequence and the lack of adult sequencing, stating that adult and larval stages have been reliably connected through DNA data, morphology, and context. The evidence described supports reliable connection through morphology, rearing, and context; the DNA component supports genus-level placement of one larva, not an adult–larva match. This distinction matters because the novelty depends on accurately attributing the life history to a named species rather than to an unidentified Phyllobaenus species. A clean fix is to revise statements of certainty so that DNA supports genus placement and does not serve as the primary bridge between life stages.

Basic barcode quality control information is not presented, and that omission increases the risk that the COI sequence will be challenged as a numt, low-quality read, or contaminant. The manuscript provides the analytic approach and distance model used for the tree, plus BLAST comparison, but it does not state the amplified fragment length, whether the sequence was translated to check for stop codons, whether chromatograms were inspected for mixed peaks, or whether independent amplification was attempted. The primer choice can be acceptable, but it increases the need to document QC explicitly. Adding a short QC paragraph in Methods and stating chromatogram availability would substantially reduce skepticism.

The phylogenetic presentation is a neighbor-joining tree built from Kimura 2-parameter distances using the ten closest BLAST hits. This construction is acceptable as an illustrative similarity figure, but it is not a strong inferential framework for taxonomic placement when the closest match is distant and sampling is restricted to BLAST neighbors. The absence of support values and broader clerid outgroups means the tree cannot robustly test alternative placements within Cleridae, and it cannot support species-level identity. Recasting the figure as a distance visualization, adding bootstrap support, and supplementing with a likelihood-based analysis with broader Hydnocerinae sampling would strengthen the molecular component without changing the biological story.

The argument that obligate parasitism or host specificity is unlikely because prevalence is very low and the ant populations are endemic compared to the beetle’s wider distribution is not logically compelled by the data. Rare prevalence is compatible with high specialization in social-insect associates, and broad geographic ranges do not exclude specialized local host use if the parasite uses different hosts across regions or if suitable conditions are patchy. The manuscript also acknowledges context dependence in other parasite systems, which makes a strong inference against obligate parasitism less consistent. A safer approach is to label facultative versus obligate status and host range as open questions, then present multiple plausible explanations for rarity, such as low encounter rate, strong colony defenses, seasonal timing, limited access stage, or sampling detection limits.

---

## [Author Response · Author response to Decision Letter 2]

27 Feb 2026

Responses to the editor

1- The adult specimen identified morphologically as Phyllobaenus obscurus was not sequenced, because extraction failed due to poor condition, and no adult COI is available for direct comparison. This leaves a critical gap in the identity chain: the adult diagnosis rests on morphology alone, and the larval barcode cannot be anchored to that adult voucher. The statement that DNA data connect adult and immature stages is not supported when the adult sequence is missing. A practical improvement is to revise the text to separate three claims: adult identification by morphology, larval placement by COI, and the inferred association based on rearing and co-occurrence.

R: We have rewritten the text to clearly separate the reliable results (morphological identification of the adult clerid beetle) from those inferred from our data and our knowledge of the host ants (identity of the immature stages and match with the adult Phyllobaenus obscurus).

2- The COI BLAST outcome reported in the Results creates a second identity vulnerability. The closest available sequence is described as only 88.74% identical, paired with a Kimura 2-parameter distance of about 0.116, which is far outside typical species-level barcode expectations for insects and can be large even for many congeneric comparisons. That pattern does not automatically invalidate genus placement, but it does mean that phrases like “placed it without any ambiguity” and “connect adult and immature stages” are overstated relative to the reported similarity metrics. A more defensible interpretation is that the COI fragment supports assignment to Cleridae and is consistent with Phyllobaenus, with species-level confirmation pending a reference sequence for P. obscurus and successful sequencing of the adult voucher.

R: Thank you for your comment. We have re-worded the corresponding parts of the results and discussion, being more cautious and conservative about the inferences that can be drawn from our data.

However, because of the different evolutionary histories of divergent taxa, it is difficult to find a fixed threshold for species/genus delimitation that is suitable to all species/genera. In the case of insects, there is high intraspecific genetic variation. For example, Zhang & Bu (2022) found that approximately one quarter (26.6%) of the species of Insecta (duly identified and with high-quality sequences n = 64,414) in BOLD Systems exhibited high intraspecific genetic variation (> 3%). For Coleoptera, this study found 8,968 species with intraspecific genetic distances greater than 3%.

But you are right, confirmation at the species level of our clerid requires a reference sequence for P. obscurus. This is beyond the scope of this study, and in the revised version, we have taken a more cautious approach in both the results and the discussion.

Zhang H, Bu W. 2022. Exploring large-scale patterns of genetic variation in the COI gene among Insecta: Implications for DNA barcoding and threshold-based species delimitation studies. Insects 13, 425). https://doi.org/10.3390/insects13050425

3- The Discussion repeats a strong linkage claim that is difficult to justify given the single larval sequence and the lack of adult sequencing, stating that adult and larval stages have been reliably connected through DNA data, morphology, and context. The evidence described supports reliable connection through morphology, rearing, and context; the DNA component supports genus-level placement of one larva, not an adult–larva match. This distinction matters because the novelty depends on accurately attributing the life history to a named species rather than to an unidentified Phyllobaenus species. A clean fix is to revise statements of certainty so that DNA supports genus placement and does not serve as the primary bridge between life stages.

R: Thank you for pointing that out. You are right, we can only say that available evidence (morphology, rearing and co-occurrence) supports the existence of a reliable link between the larva and the adult. However, in accordance with the principle of parsimony, which states that among several competing hypotheses, the simplest (the one requiring the fewest assumptions) is generally the best, it is reasonable to attribute the different life stages observed in our study to the same species of Phyllobaenus. We have modified some of our statements accordingly and toned down our assertions.

4- Basic barcode quality control information is not presented, and that omission increases the risk that the COI sequence will be challenged as a numt, low-quality read, or contaminant. The manuscript provides the analytic approach and distance model used for the tree, plus BLAST comparison, but it does not state the amplified fragment length, whether the sequence was translated to check for stop codons, whether chromatograms were inspected for mixed peaks, or whether independent amplification was attempted. The primer choice can be acceptable, but it increases the need to document QC explicitly. Adding a short QC paragraph in Methods and stating chromatogram availability would substantially reduce skepticism.

R: Thank you for pointing this out and for guiding us with your advice throughout the review process. Following this comment, we performed a FastQC analysis, and the graphs for both reads have been attached for your information (see below). The two warning flags arise from the fact that the analysis was performed on a single sequence, whereas this method is primarily designed for datasets containing multiple sequences. This shows that the sequence analyzed in this study unambiguously fits the 40 threshold criteria of nucleotide site quality. The chromatograms were examined for mixed peaks, and any ambiguities were resolved by comparing the forward and reverse sequences. We have also added to the revised text that the COI fragment was sequenced in both directions and that its translation revealed no internal stop codons (see l. 233-240 in the revised version).

5- The phylogenetic presentation is a neighbor-joining tree built from Kimura 2-parameter distances using the ten closest BLAST hits. This construction is acceptable as an illustrative similarity figure, but it is not a strong inferential framework for taxonomic placement when the closest match is distant and sampling is restricted to BLAST neighbors. The absence of support values and broader clerid outgroups means the tree cannot robustly test alternative placements within Cleridae, and it cannot support species-level identity. Recasting the figure as a distance visualization, adding bootstrap support, and supplementing with a likelihood-based analysis with broader Hydnocerinae sampling would strengthen the molecular component without changing the biological story.

R: In the previous version of the manuscript, we presented a neighbor-joining tree constructed using Kimura 2-parameter distances and limited to the ten closest BLAST hits, mainly to illustrate sequence similarity. In fact, the distances are based only on COI variation, and initial analyses performed with MEGA did not produce a topology with bootstrap values supporting the nodes.

Following your suggestion, we expanded the sampling of taxa to include additional Hydnocerinae sequences available in GenBank. The resulting phylogenetic reconstructions are shown below: Figure A shows the neighbor-joining tree, and Figure B shows the maximum likelihood tree, both analyses having been performed with 1,000 bootstrap replications to assess nodal support.

However, the bootstrap values are very low for all nodes, mainly due to the very limited number of Hydnocerinae sequences available and the quality of these sequences (we realigned some of them). Since our aim was not to produce a phylogenetic hypothesis for the subfamily (which would require sequencing several genes), but to illustrate the similarities between sequences, we prefer to keep the previous tree for illustrative purposes. We have explicitly indicated this in the figure caption.

Figure A. Neighbor-joining tree.

Figure B. Maximum likelihood tree.

6- The argument that obligate parasitism or host specificity is unlikely because prevalence is very low and the ant populations are endemic compared to the beetle’s wider distribution is not logically compelled by the data. Rare prevalence is compatible with high specialization in social-insect associates, and broad geographic ranges do not exclude specialized local host use if the parasite uses different hosts across regions or if suitable conditions are patchy. The manuscript also acknowledges context dependence in other parasite systems, which makes a strong inference against obligate parasitism less consistent. A safer approach is to label facultative versus obligate status and host range as open questions, then present multiple plausible explanations for rarity, such as low encounter rate, strong colony defenses, seasonal timing, limited access stage, or sampling detection limits.

R: Thank you for your thorough reading of our text. We have followed your suggestion. In the previous version, we left this question unresoved (see l. 411-414 of the previous version), but you are right, we discussed facultative parasitoidism in depth without addressing obligate parasitoidism. We have reworded certain parts of this paragraph to leave this as open questions needing future research endeavor (see l. 419-431 of the revised version).

(See attached file "Response to editor's comments" for the figures)

---

## [Editor Report · Decision Letter 2]

2 Mar 2026

Ant parasitoidism in checkered beetles: Phyllobaenus obscurus developing inside intact cocoons of two species of the Ectatomma ruidum species complex

PONE-D-25-54426R2

Dear Dr. Lachaud,

We’re pleased to inform you that your manuscript has been judged scientifically suitable for publication and will be formally accepted for publication once it meets all outstanding technical requirements.

Kind regards,

Petr Heneberg

Academic Editor

PLOS One

---

## [Editor Report · Acceptance letter]

PONE-D-25-54426R2

PLOS One

Dear Dr. Lachaud,

I'm pleased to inform you that your manuscript has been deemed suitable for publication in PLOS One. Congratulations! Your manuscript is now being handed over to our production team.

Kind regards,

on behalf of

Dr. Petr Heneberg

Academic Editor

PLOS One